# Religious Affiliations Influence Health-Related and General Decision Making: A Brazilian Nationwide Survey

**DOI:** 10.3390/ijerph18062873

**Published:** 2021-03-11

**Authors:** Marcelo Borges, Giancarlo Lucchetti, Frederico C. Leão, Homero Vallada, Mario F. P. Peres

**Affiliations:** 1PROSER, Programa de Espiritualidade e Religiosidade, Instituto de Psiquiatra, Hospital das Clinicas HCFMUSP, Faculdade de Medicina, Universidade de Sao Paulo, Sao Paulo 05403-903, Brazil; leaofc@gmail.com (F.C.L.); hvallada@usp.br (H.V.); mariop3r3s@gmail.com (M.F.P.P.); 2School of Medicine, Federal University of Juiz de Fora, Juiz de Fora 36038-330, Brazil; g.lucchetti@yahoo.com.br; 3Departamento & Instituto de Psiquiatria (LIM-23) HCFMUSP, Sao Paulo 05403-903, Brazil; 4Faculdade de Medicina FMUSP, Universidade de Sao Paulo, Sao Paulo 05403-903, Brazil; 5Hospital Israelita Albert Einstein, São Paulo 05403-903, Brazil

**Keywords:** nationwide, online survey, religion, sensitive information, public health

## Abstract

Approximately 90% of the world’s population is involved in some spiritual/religious practice, and this dimension has a relevant role in life. Many studies demonstrate the associations between spirituality/religiosity (S/R), and physical, mental, and social health. Systematic reviews have indicated positive associations; however, the mechanisms behind religious coping are not fully understood. The present study aimed to examine the role of religious affiliation in general (ordinary) and health-related decisions. A nationwide, population-based, cross-sectional study was conducted in Brazil using a self-administered online survey. How much religious affiliation influences decision making was investigated. A total of 1133 participants were included, who were classified as Catholics (43.9%), Evangelicals (18.7%), spiritualists (12.8%), non-religious (11.9%), and others (12.7%). Most participants (66.5%) believed that their religious affiliations had moderate to high influences on their decisions. Participants rated the influence as high in marriage (62.7%), in donations (60.1%), in volunteering (55%), in friendships (53.9%), and in work (50.5%). Concerning health-related decisions, the influence was rated as high in drug use (45.2%), in accepting medical recommendations (45%), and in smoking (43.2%). The influence of religious affiliation on general decision making was significantly correlated for dietary restrictions (*r* = 0.462), alcohol consumption (*r* = 0.458), drug use (*r* = 0.441), tobacco consumption (*r* = 0.456), and refusal of medical recommendations (*r* = 0.314). Improving the understanding of how a patient’s beliefs, practices, and experiences affect their health may help healthcare practitioners to take into account religious considerations, not only regarding influences on habits but also regarding adherence to medical treatment advice.

## 1. Introduction

Approximately 90% of the world’s population is involved in some spiritual/religious practice, and this dimension plays a relevant role in life [1]. Evidence of spirituality and religiousness’s effect on health has been constantly growing. Approximately 30,000 articles were published in the last 15 years in this field of research according to a bibliometric analysis [2].

Many studies have demonstrated associations between spirituality/religiosity (S/R) and physical, mental, and social health [2]. Systematic reviews have shown that most studies indicated some positive association in which the S/R was a protective factor, but some studies found negative or neutral associations [3,4]. Many professional organizations recognize that spiritual care is an important component of healthcare, such the World Health Organization (WHO). The Joint Commission on the Accreditation of Healthcare Organizations (JCAHO) [5] recommended including S/R in clinical care and education. Concerning mental health, the World Psychiatric Association, American Psychological Association [6], and Royal College of Psychiatrists [7] have sections dedicated to S/R issues. The way patients experience the S/R dimension can be either protective or harmful, depending on how these resources are used in the health/disease process (religious coping). The mechanisms behind religious coping are not fully understood, but religion is a relevant aspect of an individual’s cultural background, important not only in daily decisions (e.g., dating, friendship, and clothing choices) but also health-related attitudes (e.g., diet, smoking, and drug use). Beliefs, practices, and experiences can influence decisions involving medical treatment. Religiousness is associated with adherence, giving meaning to an illness [8], and care to one’s body [9]. Religiousness can be associated with acceptance, refusal, or rejection of medical recommendations [10]. Religions significantly influence important health habits (use/abuse of alcohol [11], drugs, and smoking [12]). Since patient care is influenced by cultural and religious beliefs, assessing how much a particular religious affiliation influences decision making could contribute to a comprehensive understanding of the amount or the magnitude of this relationship.

On the one hand, studies in this field have not been designed to consider patients’ perceptions about religion [13], and the basis may start with data being extracted from nationwide censuses not primarily designed to assess religion’s scientific aspects in a broader way [14]. On the other hand, most studies were conducted in specific populations [15]. Sometimes, religious affiliation was compared with non-religiousness [16]. The majority of studies have been carried out in the United States and maybe are not fully applicable in other contexts and cultures [17]. Previous international studies have already used online surveys to investigate spirituality and religion in the general population [18,19], and the online approach is still mostly used for health professionals [20,21]. On the specific subject of religious affiliation, no study was found that investigates the patient’s perception of its influence on daily or health-related decision making.

Therefore, this study aimed to evaluate how much religious affiliation influences daily life and health-related decisions, thereby providing to healthcare providers more information to address religious considerations and demonstrate how a patient’s beliefs, practices, and experiences may affect his/her health.

## 2. Materials and Methods

This is a nationwide, cross-sectional, quota sampled, self-administrated online survey study. Invitations were sent by Qualtrics^®^ Panels, one of the main companies offering online instrument services [22], for the project “Spiritual and Religious Beliefs, Practices and Experiences in the General Population.” It was sponsored by the Interfaith Coalition on Spirituality and Health (coalizaointerfe.org, accessed on 10 September 2020), a Brazilian institution composed of representative members of several religious affiliations, and non-religious faith and professional healthcare practices in Brazil. Data were collected between 28 June 2016 and 22 August 2016, through a self-administered online survey. Respondents were invited to participate and complete the online survey, and those who agreed signed an online consent term.

The sample size was determined by non-probability quota samples, thereby assuring that distinct subgroups of the population and their characteristics were represented. Quotas were set to limit the respondents according to social class distribution, age, gender, and geographic location, so the population surveyed could meet the same profile of the general adult population in Brazil, according to the 2010 census. A previous article [14] showed that using this sampling procedure could result in accurate representative surveys for Brazil as a whole. Inclusion criteria were 18 years old or more, and online access to a website, email, or social media network. Missing background data, invalid email addresses, incomplete answers, and missing responses were exclusion criteria. The survey took less than 30 min and attention filters and quality check questions were used.

A question about religious affiliation was asked: “What is the best option below to define your current religious affiliation?” Response options included 16 options: “Catholic,” “Evangelical,” “Kardec Spiritist,” “Jewish,” “Buddhist,” “Umbanda,” “Candomble,” “atheist,” “agnostic,” “spiritualist,” “Christian,” “Jehovah’s Witness,” “Seicho-no-ie,” “Wicca,” “no religion,” and “others.” Due to the limited number of responders in some affiliations, we re-allocated the sample into 6 groups: “Catholics,” “Evangelicals,” “spiritualists” (Kardec spiritists and spiritualists), “other Christians,” “non-religious” (agnostic, atheist, and no religion), and “other denominations” (Jewish, Buddhist, Umbanda, Candomble, Jehovah’s Witness, Seicho-no-ie, Wicca, and others). Kardec Spiritists and spiritualists supposedly behave similarly. Although “others” seems heterogeneous, their sample size was still small, justifying their being analyzed together.

The self-reported influence of religious affiliation on general life was assessed by the question: “How much does religious affiliation influence your decisions?” The self-reported influence of religious affiliation on various attitudes was examined by: “How much does your religious affiliation influence your”: “dating,” “marriage,” “friendships,” “work,” “clothing choices,” “volunteering,” “donations,” and “politics.” The self-reported influence of religious affiliation on health-related attitudes was examined by: “How much does your religious affiliation influence” your decisions regarding “diet,” “alcohol consumption,” “drug use,” “tobacco consumption,” “refusal of medical recommendations,” and “acceptance of medical recommendations”?

All questions presented 5 alternatives: 1 (very little), 2 (a little bit), 3 (moderately), 4 (quite a lot), and 5 (a lot). Analysis was performed with 95% confidence intervals. For correlations, we used Pearson tests; for comparisons, paired *t*-tests for normal samples. The Wilcoxon test was used for non-parametric comparisons, and Fisher and chi-square tests were used to compare proportions (IBM SPSS Statistics version 25). The study followed international ethical regulations (Declaration of Helsinki principles) and was approved by the local Ethics Research Committee of IPq HCFMUSP (CAAE 64956717.4.0000.0065). All participants gave informed consent online.

## 3. Results

From 1252 participants, 1131 (90.3%) completed the selected questions. The sample was composed of 536 (47.4%) men and 595 (52.6%) women, with a mean age of 40.5; 57.9% of the participants were married and 37.8% had a monthly family income between US800.00 and US2000.00 (Table 1).

The sample was categorized into six religious groups: Catholics (43.9%), Evangelicals (18.7%), spiritualists (12.8%), other Christians (6.6%), other denominations (6.1%), and non-religious (11.9%). Regarding how much their religious affiliations influenced their decision making, 21.4% of the participants reported “very little,” 12.1% “a little bit,” 23.2% “moderately,” 26.1% “quite a lot,” and 17.2% reported “a lot.”

Religious affiliation correlated with daily decisions to various degrees: “dating” (*r* = 0.530), “marriage” (*r* = 0.551), “friendships” (*r* = 0.565), “job” (*r* = 0.571), “clothing choices” (*r* = 0.507), “volunteering” (*r* = 0.568), “donations” (*r* = 0.590), and “politics” (*r* = 0.349). Likewise, health-related decisions correlated as follows: “diet” (*r* = 0.462), “alcohol consumption” (*r* = 0.458), “drug use” (*r* = 0.441), and “tobacco consumption” (*r* = 0.456). “Acceptance of medical recommendations” has a correlation of 0.560. “Refusal of medical recommendations” presented as *r* = 0.314, demonstrating that the influences of religious affiliation were lower on health-related aspects, as detailed by religious affiliation categories (Table 2).

Most participants’ (66.5%) believe that their religious affiliations have moderate to high influences on their general decisions. Participants rated the influence of religious affiliation as high on marriage (62.7%), on donations (60.1%), on volunteering (55%), on friendship (53.9%), on work (50.5%), on dating (49.6%), on choice of clothing (42%) and on politics (27.2%) (Figure 1).

Concerning health-related decisions, the influence of religious affiliation was high on drug use (45.2%), on accepting medical recommendations (45%), on smoking (43.2%), on alcohol consumption (39.7%), on dietary restrictions (37.1%), and on refusing medical advice (22.6%) (Figure 2).

Evangelicals had the highest levels of religious influence within daily life and health-related decisions, and as expected, the non-religious group had the least influence from religion (though they were referring to the influences of beliefs on life decisions).

## 4. Discussion

Our data show religious affiliation influences daily life and health-related decisions. Overall, the influence of religious affiliation on general decisions was considered high by the participants, particularly in areas such as marriage, donations, friendships, and work. However, this influence is also considerable in health-related decisions, such as the refusal of medical advice, diet, smoking, and alcohol use. These findings provide further support to the scientific literature, which has been showing that religion is a relevant aspect of individuals’ cultural background [1], important not only in daily life but also in health-related decisions and attitudes [2]. Additionally, the way patients experience the S/R dimension can be either protective or harmful, depending on how these resources are used [23,24].

Historically, religions incorporated forms of fasting and self-starvation. Many important religious figures, such as Moses, Buddha, and Jesus abstained from eating as part of ascetic/mystic practices. A controlled trial study found evidence of treatment via spirituality-oriented groups based on the eating attitude test (EAT), and eating disorder specific symptoms, such as binging, purging, restricting, and attitudes about food and dieting [25]. That was corroborated in a previous correlational study [26]. In our study, 63% of participants reported a small or very small influence of religion on diet, as opposed to 37% reporting a moderate influence or above. Although the majority said religion does not affect their dietary patterns, it is very important to consider that one-third of the general population says the opposite. Several health problems are sensitive to dietary patterns—obesity, hypertension, diabetes, and dyslipidemia, not to mention less prevalent disorders. Clinicians should be aware of a patient’s religiosity so a treatment plan could be tailored and would not be compromised by their adherence to dietary recommendations.

Studies have demonstrated a relationship between abstinence and alcohol and use/abuse [27], but also pointed out the difficulty of identifying which factors of the spiritual dimension are present in this mediation. Increased S/R involvement is related to lower alcohol consumption and better outcomes in alcohol abuse treatments. Although religious affiliation does not define alcohol-related attitudes, it may contribute to abuse [28]. Attitudes toward alcohol consumption in this study showed 39.7% of individuals rating religious influence as moderate and above. This finding supports the idea that religion is a mediator, suppressing alcohol consumption. One may hypothesize that this is due to the social conduct rules of one’s religion, but it could be related to spiritual aspects too. Concerning drug use, an epidemiological study conducted in the U.S. presented a higher risk of having started the use of cannabis, cocaine, or other extra-medical drugs for the non-religious. Additionally, those who declared less importance for religion in life were more likely to use drugs [29]. A causal-comparative drug abuse study demonstrated neutral effects of religion on addiction; however, the sample had high religiosity and commitment to beliefs [30]. Studies with minorities showed religiosity to be a protective factor [31]. Regarding the influence on drug use, this study demonstrated 45.2% of participants rating it as moderately important or more so.

For tobacco consumption, a nationwide study performed in the United Kingdom [12] indicated that the prevalence of tobacco use is higher among people who identified themselves to be without religion (66.2%) than those who declared themselves to be Christians (60.0%) and relatively lower among those who declared themselves to be Muslims (35.2%). In this study, only 9.6% of non-religious people rated their beliefs as having an influence of moderate and above that; 57.3% of Christians and 62.1% of Evangelicals declared an influence of moderate or above. However, the mechanisms involved in this process still require further study. Among the possible reasons is the fact that many religious denominations prohibit the use of substances such as tobacco; the time spent in activities related to the S/R dimension has decreased risk behaviors; religious affiliation can increase the likelihood of confrontations regarding one’s character; and participation in a religious institution has the characteristic of social involvement, which can discourage such behaviors [32].

Concerning the influence of religion on following or refusing medical advice, one study showed trust in physicians and the health system. A nationwide U.S. study showed a positive association with adherence to treatment and strength of religious affiliation [33]. A case series study investigated patients, noting that the S/R dimension helped adherence to treatment, indicating a positive association [9]. Another nationwide study with 41-year-olds and older showed that high religious attendance (more than a weekly) granted a 40% reduced hazard of mortality compared with those who never attended, and there was a positive association between religious attendance and the importance of religion to the respondent. However, higher levels of religiousness are by no means always associated with better health. Our findings show amounts of influence of “very little,” (40.5%), “a little bit” (14.5%), “moderately” (22.2%), “quite a lot” (15.1%), and “a lot” (7.7%) on accepting medical recommendations. However, for denying or refusal of medical recommendations, the results were: “very little” (63.2%), “a little bit” (14.1%), “moderately” (13.0%), “quite a lot” (5.7%), and “a lot” (4.0%).

Thus, awareness about the global role of S/R in health has been increasing [34]. However, nationwide studies frequently do not have a primary design to assess religious issues as a main topic [14]. This study demonstrated how religious affiliation influences attitudes toward daily life and health-related decisions. It may have clinical relevance for healthcare professionals. A good approach needs awareness of religions to avoid unhelpful judgment and preconceptions [35], to increase trust with each patient.

This study had some limitations that should be considered. First, this was a cross-sectional survey, avoiding inferences about cause and effect. Second, data collection was carried out only for persons 18 years old or older who had online access to email, social media, and websites; it is possible that some groups of the population, such as residents from rural areas, those with lower socioeconomic levels, and older adults, were included at a lower frequency. Finally, we used self-reported measures to investigate the influences of religious affiliation on daily decisions, instead of evaluating their behavior in real-life scenarios. However, previous international studies have already used online surveys to investigate spirituality and religion in the general population [18,19] and the online approach is still mostly used by health professionals [20,21].

## 5. Conclusions

In conclusion, religious affiliation influences daily life and health-related decisions to different degrees. Healthcare providers need to consider religious aspects of patients’ lives to consider how a patient’s beliefs, practices, and experiences may affect their health, not only because of religion’s influences on habits such as diet, smoking, drinking, and drug use, but also regarding adherence to medical advice.

## Figures and Tables

**Figure 1 ijerph-18-02873-f001:**
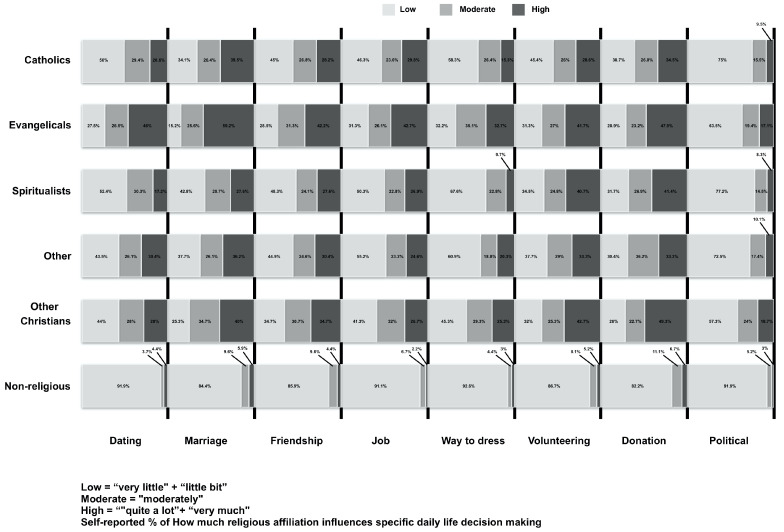
Religious affiliation’s influences in daily decisions.

**Figure 2 ijerph-18-02873-f002:**
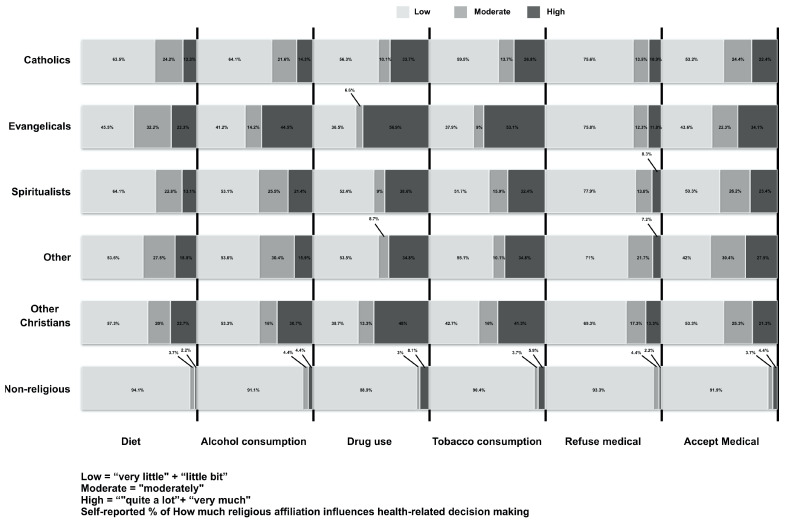
Religious affiliation’s influences in health-related decisions.

**Table 1 ijerph-18-02873-t001:** Descriptive characteristics of the sample.

	Categories	Catholics	Evangelicals	Spiritualists	Other	Other Christians	Non-Religious
Variables		Total (n = 496) n(%)	Total (n = 411) n(%)	Total (n = 145) n(%)	Total (n = 69) n(%)	Total (n = 75) n(%)	Total (n = 135) n(%)
**Gender**	Female	235 (47.4)	122 (57.8)	94 (64.8)	36 (56.2)	45 (60)	63 (43.7)
	Male	261 (52.6)	89 (42.2)	51 (35.2)	33 (47.8)	30 (40.0)	72 (53.3)
**Age Range (years)**	18–25	116 (23.4)	48 (22.7)	28 (19.3)	10 (14.5)	24 (32.0)	30 (22.2)
	26–35	93 (18.8)	48 (22.7)	31 (21.4)	14 (20.3)	18 (24.0)	27 (20.0)
	36–45	103 (20.8)	46 (21.8)	37 (25.5)	16 (23.2)	8 (10.7)	24 (17.8)
	46–55	80 (16.2)	22 (10.4)	23 (15.9)	18 (26.1)	12 (16.0)	22 (16.3)
	56–65	72 (14.5)	34 (16.1)	14 (9.7)	7 (10.1)	9 (12.0)	18 (13.3)
	>65	31 (6.3)	13 (6.2)	12 (8.3)	4 (5.8)	4 (5.3)	14 (10.4)
**Educational Level**	Incomplete high school	18 (3.6)	16 (7.6)	5 (3.4)	9 (13)	2 (2.7)	7 (5.2)
	Completed high school	192 (38.7)	105 (49.8)	41 (28.3)	33 (47.8)	29 (38.7)	52 (38.5)
	Completed university	193 (38.9)	61 (28.9)	60 (41.4)	19 (27.5)	30 (40.0)	62 (45.9)
	Postgraduated	93 (18.8)	29 (13.7)	39 (26.9)	8 (11.6)	14 (18.7)	14 (10.4)
**Marital Status**	Single	149 (30.0)	56 (26.5)	43 (29.7)	24 (34.8)	29 (38.7)	74 (54.8)
	Judicially separated	12 (2.4)	6 (2.8)	3 (2.1)	0 (0)	0 (0)	4 (3.0)
	Divorced	28 (5.6)	7 (3.3)	6 (4.1)	6 (8.7)	3 (4.0)	7 (5.2)
	Married	300 (60.5)	138 (65.4)	88 (60.7)	38 (55.1)	43 (57.3)	48 (35.6)
	Widowed	7 (1.4)	4 (1.9)	5 (3.4)	1 (1.4)	0 (0)	2 (1.5)
**Monthly income range (USD) ***	>4114	46 (10.6)	14 (8.1)	12 (10.2)	5 (9.3)	5 (7.7)	8 (6.8)
	2058–4114	132 (30.5)	36 (20.8)	36 (30.5)	10 (18.5)	14 (21.5)	31 (26.5)
	823–2057	158 (36.5)	61 (35.3)	48 (40.7)	18 (33.3)	28 (43.1)	50 (42.7)
	411–822	71 (16.4)	40 (23.1)	16 (13.6)	17 (31.5)	10 (15.4)	20 (17.1)
	<411	26 (6.0)	22 (12.7)	6 (5.1)	4 (7.4)	8 (12.3)	8 (6.8)
**Race/Color**	White	325 (65.5)	107 (50.7)	95 (65.5)	38 (55.1)	34 (45.3)	94 (69.6)
	Black	19 (3.8)	15 (7.1)	9 (6.2)	4 (5.8)	3 (4.0)	11 (8.1)
	Asian	12 (2.4)	6 (2.8)	2 (1.4)	3 (4.3)	2 (2.7)	4 (3.0)
	Pardo (mixed)	138 (27.8)	81 (38.4)	38 (26.2)	22 (31.9)	36 (48)	25 (18.5)
	Indigenous	2 (0.4)	2 (0.9)	1 (0.7)	2 (2.9)	0 (0)	1 (0.7)
**Region**	Northeast	144 (29)	51 (24.2)	44 (30.3)	18 (26.1)	18 (24)	33 (24.4)
	South	66 (13.3)	34 (16.1)	23 (15.9)	11 (15.9)	9 (12)	18 (13.3)
	Midwest	37 (7.5)	13 (6.2)	10 (6.9)	4 (5.8)	9 (12)	16 (11.9)
	Southeast	213 (42.9)	93 (44.1)	62 (42.8)	27 (39.1)	35 (46.7)	60 (44.4)
	North	36 (7.3)	20 (9.5)	6 (4.1)	9 (13)	4 (5.3)	8 (5.9)

* Conversion from Brazilian Real using the mid-price for the dollar in July 2018 (1 USD = BRL 3.83).

**Table 2 ijerph-18-02873-t002:** Correlations between generic questions, daily decisions, and health-related questions.

	Catholics	Evangelicals	Spiritualists	Other	Other Christians	Non Religious
**Dating**	**0.50 ***	**0.41 ***	**0.36 ***	**0.55 ***	**0.49 ***	**0.34 ***
**Marriage**	**0.51 ***	**0.44 ***	**0.39 ***	**0.46 ***	**0.53 ***	**0.37 ***
**Friendship**	**0.59 ***	**0.41 ***	**0.37 ***	**0.48 ***	**0.43 ***	**0.50 ***
**Job**	**0.62 ***	**0.34 ***	**0.41 ***	**0.42 ***	**0.48 ***	**0.55 ***
**Dress**	**0.50 ***	**0.40 ***	**0.26 ***	**0.44 ***	**0.46 ***	**0.60 ***
**Volunteers**	**0.57 ***	**0.34 ***	**0.43 ***	**0.61 ***	**0.64 ***	**0.52 ***
**Donation**	**0.62 ***	**0.37 ***	**0.43 ***	**0.56 ***	**0.56 ***	**0.52 ***
**Politics**	**0.35 ***	**0.22 ***	**0.29 ***	**0.20 ***	**0.44 ***	**0.30 ***
**Diet**	**0.47 ***	**0.32 ***	**0.32 ***	**0.49 ***	**0.32 ***	**0.53 ***
**Alcohol**	**0.39 ***	**0.34 ***	**0.43 ***	**0.52 ***	**0.34 ***	**0.44 ***
**Drug use**	**0.38 ***	**0.29 ***	**0.40 ***	**0.37 ***	**0.45 ***	**0.33 ***
**Tobacco**	**0.36 ***	**0.36 ***	**0.49 ***	**0.43 ***	**0.50 ***	**0.33 ***
**Refuse Medical**	**0.30 ***	**0.19 ***	**0.29 ***	**0.27 ****	**0.33 ***	**0.54 ***
**Acept Medical**	**0.61 ***	**0.31 ***	**0.50 ***	**0.59 ***	**0.46 ***	**0.50 ***

* Pearson’s correlation is significant at the 0.01 level (2-tailed); ** Pearson’s correlation is significant at the 0.01 level (1-tailed). Correlation Level: High, 

; Medium, 

; Weak, 

; Very week, 

.

## Data Availability

Project “Spiritual and Religious Beliefs, Practices and Experiences in the General Population,” was sponsored by Interfaith Coalition on Spirituality and Health (coalizaointerfe.org, accessed on 10 September 2020), a Brazilian institution composed of representative members of several religious affiliations, and non-religious faith and healthcare practices in Brazil.

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
