# Peer review of "Religious Affiliations Influence Health-Related and General Decision Making: A Brazilian Nationwide Survey"

_ijerph, 2021, doi:10.3390/ijerph18062873_

Round 1

Reviewer 1 Report

The aim to assess a dimension of Rel/Spir that might be part of the overal relationships between Rel/Spir and health is a very valuable goal. yes, assessing religious identification and its influence is also very valuable and much called for especially in countries other than the USA which is always being studied...

I have some questions about this effort:

  1. More detail about the 'panel' to whom invitations were sent is in order. How does it relate to the population? How is this ensured? I realise that this is becoming an accepted way of proceeding and in fact have used it myself, but always with serious concerns. For example different panels have yielded different results. At least the tone of reporting needs to be more qualified.
  2. How were the larger frame of response categories for religious identity reduced to the smaller number used in analysis? This needs to be made more explicit.
  3. Assessments of impact are rated moderate etc. Against what comparators. What/who sets the benchmark here??/
  4. I really place very limited stock in 'self-assessments' in general and hardly at all in an area like this. I want to see behavioural measures that differ by rel ident. This weakness makes the paper hardly any use at all. Do people from identity group A differ in measures of health behaviour - smoking, drug taking, abortion etc etc in fact, not just in their minds. 

Reviewer 2 Report

The article deals with an important and updated topic. It presents interesting results and due to the wide, multidisciplinary interest in the role of religion and its influences it can be found as valuable contribution by representatives of many different disciplines of science. It raises though a few fundamental issues that need to be seriously taken into consideration. First in the theoretical introduction it does not relate to existing approaches and findings referring only to the US in a very general, and not sufficient way. It leaves thus the impression the theoretical framework is very poor and somehow omitted by the authors. Secondly the basic methodological critics refers to the fact that the data presented was collected in 2016 which means almost 5 years ago. This makes the research quite outdated if we consider the dynamics of religious changes and fluctuations in the religious attitudes. It is necessary that the authors explain this issue and convincingly argue about that. Finally the article contains quite a few language mistakes which make it sometimes difficult to understand presented contents.

Reviewer 3 Report

P.T. Authors, thank you for the research and the article. I have same small comments, the validity of which can be judged only by you. First, shouldn't a group defined as "Christians" be renamed as "other Christians"? Catholics and Evangelicals are Christians, so either the name in the research includes them as well, or the name should be changed. Secondly, religions have different messages regarding everyday life customs, ethics, or, for example, diet. Catholics do not attach big importance to clothing, they are allowed to consume alcohol in moderation, Jews have specific food regulations, Muslims too, Islam has rules for women's dress, Jehovah's Witnesses do not allow blood transfusion, etc. Even in the case of vaccination against COVID-19 the fact that in the production of vaccines abortion-derived material was used in some Churches required the development of a separate ethical position on the permissibility of their use. It would be worth confronting the obtained declarations also with the content of the professed faith. For this, you would probably need some theologians on the team. Moreover, to study the relationship between the declared strength of faith and health practices consistent with the indications of a given religion would be interesting. However, I do not know if you have the necessary data to complete your analyzes in this way. 

Round 2

Reviewer 1 Report

Authors have worked to improve this article and it is improved